

# Prevalence of obesity and associated risk factors among university students using the newly developed Student Lifestyle and Obesity Risk Questionnaire (SLORQ): a cross-sectional study

Mohamed K. Seyam

Department of Physical Therapy and Health Rehabilitation, College of Applied Medical Sciences, Majmaah University, Majmaah, Riyadh, Saudi Arabia

## ABSTRACT

**Background**. Obesity is a significant global health challenge, with a rising prevalence among young adults. University students are particularly vulnerable to lifestyle transitions and environmental influences. However, current assessment tools lack comprehensive multidimensional coverage of obesity risk factors, especially tailored to the cultural context of Saudi University students, representing a significant gap this study aims to address.

**Objectives**. To determine the prevalence of obesity among university students and identify behavioral, environmental, and demographic risk factors contributing to obesity *via* the Student Lifestyle and Obesity Risk Questionnaire (SLORQ).

**Methods**. This cross-sectional study was conducted from March to July 2024 and involved 907 students across 11 colleges at Majmaah University. Anthropometric data (height and weight) were objectively measured by the primary author, with students barefoot and wearing minimal clothing. The participants completed a self-reported validated, bilingual questionnaire (SLORQ) assessing six domains: physical activity, dietary habits, sleep patterns, weight management, metabolic health, and environmental factors. Data were analyzed using descriptive statistics (means, standard deviations), chi-square tests for categorical associations, and multiple linear regression for continuous predictors. Structural validity was assessed *via* principal component analysis (PCA); reliability included Cronbach's $\alpha$ ($\geq 0.70$) and intrarater reliability, intraclass correlation coefficients (ICC).

**Results**. Following data collection, statistical analyses revealed that the prevalence of obesity was 15% ($n = 136$), with 40% classified as overweight. SLORQ scores revealed inverse relationships between body mass index (BMI) and health behaviors, particularly dietary habits ($r = -0.912$, $p < 0.001$) and physical activity ($r = -0.614$, $p < 0.001$). No significant differences in BMI were observed based on gender, age, or region. Regression analysis indicated that diet, sleep, and physical activity accounted for most of the variance in BMI, highlighting their importance in obesity prevention. The questionnaire's internal consistency (Cronbach's $\alpha = 0.915$) and intrarater reliability (ICC = 0.999) were assessed to ensure validity.

**Conclusion**. Obesity among university students at Majmaah University is influenced by poor dietary practices, inadequate physical activity, and insufficient sleep. These

Corresponding author
Mohamed K. Seyam,
m.seyam@mu.edu.sa

findings underscore the need for targeted health promotion campaigns and campus-wide interventions to address modifiable risk factors. The SLORQ has proven effective in assessing multidimensional obesity risk factors and can guide future research and policymaking.

# INTRODUCTION

Obesity is a significant global health challenge that affects populations across all age groups and socioeconomic levels. Over the past several decades, the prevalence of obesity has increased, with over 650 million adults currently classified as obese (*Haththotuwa, Wijeyaratne & Senarath, 2020*). The World Health Organization (WHO) declared obesity a global pandemic, with a 2.5-fold increase in death and disability attributed to high body mass index (BMI) (*Zhou et al., 2024*).

Recent studies estimate that by 2040, over 2.26 million new cases of type 2 diabetes mellitus, chronic liver diseases, and liver cancer in South America will be attributed to obesity in young adults (*Coker et al., 2022*). Within this context, university students represent a critical demographic, as their lifestyle habits and health behaviors during this formative stage often persist into adulthood, shaping long-term health outcomes (*Lawrence, Mollborn & Hummer, 2017*). A study across 22 universities in low-, middle-, and emerging economy countries found obesity prevalence rates of 5.2% (females) and 5.5% (males) (*Peltzer et al., 2014*). A global analysis reported a 9% obesity prevalence among medical students (*Shafiee et al., 2024*). The prevalence of obesity in the Kingdom of Saudi Arabia (SA) surpasses global trends, ranging from 19.4% to 39% among adolescents and adults, respectively (*Al-Omar et al., 2024*). Additionally, the economic burden of obesity in SA amounted to $3.8 billion, representing 4.3% of the nation's total health expenditures (*Malkin et al., 2022*). However, in SA, a 15% prevalence of obesity has been reported among college students (*Alqassimi et al., 2024*).

Obesity among college students is a multifaceted issue influenced by a complex interplay of behavioral (*Desai et al., 2008*), environmental (*Jia, 2021*), psychological (*Mohd-Sidik, Lekhraj & Foo, 2021*), and biological factors (*Loos & Yeo, 2022*). Students often adopt unhealthy dietary patterns, reduce physical activity, experience irregular sleep, and face increased academic and social stress (*Oftedal et al., 2024*). These factors, combined with metabolic predispositions and environmental influences such as limited access to healthy food options and supportive facilities, significantly increase the risk of obesity (*Górczyńska-Kosiorz, Kosiorz & Dzikegielewska-Gkesiak, 2024*). Among university students, obesity has been associated with impaired cognitive function, reduced academic performance, and lower self-esteem (*Pearson et al., 2012*; *Joseph et al., 2014*; *Phelan et al., 2015*). Despite numerous available questionnaires assessing obesity risks, existing tools lack comprehensive coverage of interconnected behavioral, environmental, and cultural factors specifically

relevant to Saudi Arabian university students. Existing questionnaires may not fully capture the unique dietary, physical activity, and lifestyle patterns prevalent in this population, highlighting the need for the development of the Saudi Lifestyle Obesity Risk Questionnaire (SLORQ). Furthermore, most instruments fail to provide validated Arabic translations or culturally adapted measures. Therefore, the development and validation of a culturally sensitive, comprehensive, multidimensional tool like the SLORQ is critical.

Therefore, this study aims to (1) assess the prevalence of obesity and associated behavioral, demographic, and environmental risk factors among university students, and (2) introduce and validate a newly developed questionnaire, the SLORQ. This study contributes to the development of targeted interventions and public health strategies by providing evidence on the key factors contributing to obesity in the Saudi population, and by offering a culturally relevant tool (SLORQ) to guide future obesity prevention programs, policy-making, and health promotion initiatives.

## MATERIALS & METHODS

### Study design and study population

A cross-sectional survey was conducted among a sample of 907 (52.1% male and 47.9% female) students aged above 18 years between March 21 and July 6, 2024, from colleges at Majmaah University in SA. This study received responses from students across 11 colleges out of a total of 13 colleges at Majmaah University. The colleges included were the College of Business Administration in Majmaah, the College of Dentistry at Zulfi, the College of Education, the College of Engineering, the College of Medicine, the College of Nursing, the College of Sciences, the College of Applied Medical Sciences, the College of Sharia and Law, and the Computer Sciences and Information Technology College. The participants were selected *via* convenience sampling. The students completed a self-administered questionnaire, and their anthropometric parameters were recorded by the author of this study (Fig. 1).

### Inclusion and exclusion criteria

Male and female students above the age of 18, studying at Majmaah University. They were deemed eligible for inclusion if they had a clear understanding of either the Arabic or English language, were willing to participate in the research, agreed to provide informed consent, were willing to complete the questionnaire, and were willing to provide their anthropometric readings. Students were excluded if they were younger than 18 years, not currently enrolled at Majmaah University, unable to read Arabic or English well enough to understand the study materials, unwilling or unable to give written informed consent, declined to complete or failed to finish the questionnaire, refused to provide or permit collection of anthropometric measurements, reported any chronic medical condition likely to affect weight, metabolism, or overall health (*e.g.*, diabetes, thyroid disorders, cardiovascular disease), were absent for all data-collection sessions, had already participated in the pilot phase, or had a medical, cognitive, or psychological issue that, in the investigators' judgment, would prevent safe or reliable participation.

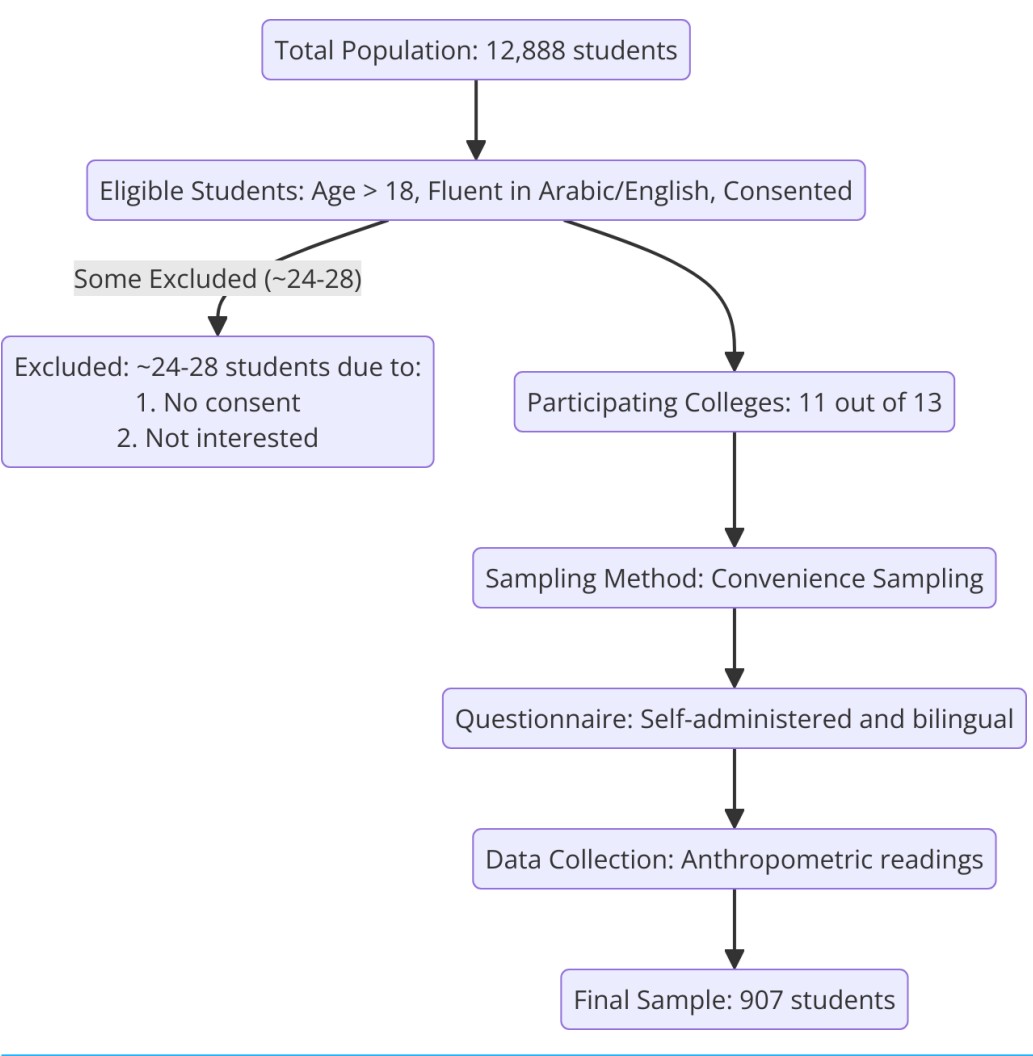

**Figure 1  Participants enrollment flow chart.**

## Development, translation and validation of SLORQ

The SLORQ was developed through a rigorous Delphi process involving an expert panel of five experienced physicians and five physiotherapists specializing in obesity research (*Okoli & Pawlowski, 2004*). The initial draft of the questionnaire was developed by the first author and subsequently reviewed for content validation. The Delphi process involved three iterative rounds in which the panel evaluated the relevance, clarity, and completeness of the items. Panel members independently rated each item on a 5-point Likert scale and provided qualitative feedback. Revisions were made based on the basis of consensus, defined as 80% agreement among panel members. The finalized questionnaire comprised eight demographic items and 24 items distributed across six domains: physical activity (five items, score range 0–17), dietary habits (six items, score range 0–20), sleep patterns (four items, score range 0–12), weight management (three items, score range 0–11), metabolic health (two items, score range 0–3), and environmental and family factors (four items,

score range 0–15). The total possible score ranges from 0 to 78, with higher scores reflecting healthier behaviors and lower obesity risk.

The questionnaire was presented in a dual-language format (English and Arabic) to ensure accessibility for all participants. The translation process adhered to Beaton's guidelines for cross-cultural adaptation. Initially, a bilingual professional translator with expertise in medical terminology translated the English version into Arabic. A bilingual study author reviewed the forward translation for linguistic and cultural appropriateness, resolving any inconsistencies with the translator. The Arabic version was then independently back-translated into English by another bilingual expert who was blinded to the original questionnaire. Both versions underwent a thorough review by the research team to ensure semantic, idiomatic, and conceptual equivalence between the languages. The dual-language version was pilot-tested on a focus group of ten students to assess clarity, ease of use, and completion time, with their feedback incorporated into the final version. The final dual-language questionnaire, validated for cultural and linguistic appropriateness to assess obesity risk factors and health behaviors among students at Majmaah University (Fig. 2).

## Outcome measures

The finalized questionnaire comprises 24 items divided into six domains designed to comprehensively assess obesity risk factors, health behaviors, and environmental influences. The questionnaire is bilingual and available in both English and Arabic to ensure accessibility for all participants at Majmaah University. Demographic information and data on age, sex, academic level, region, and anthropometric measurements (height, weight, BMI) were collected. The first domain, physical activity, evaluates the type, frequency, and intensity of physical activities, as well as campus-related physical activity participation, contributing a maximum score of 17 points. It captures energy expenditure by assessing how different activities, from structured exercises to everyday tasks like walking and housework, vary in terms of intensity and frequency, with higher scores indicating more frequent, intense, and diverse physical activity patterns.

The dietary habits domain (maximum score: 20) examines the daily consumption of fruits and vegetables, protein sources, frequency of fast food and sugary food intake, and calorie balance.

The sleep patterns domain (maximum score: 12) assesses sleep duration, quality, and difficulty falling or staying asleep, with higher scores reflecting healthier sleep behaviors.

The weight management domain (maximum score: 11) focuses on weight stability, regular monitoring of weight and waist circumference, past weight loss interventions, and social support for healthy behaviors.

The final domain, environmental and family factors (maximum score: 15), evaluates the availability of healthy food options and campus amenities, awareness of nutritional content, family history of metabolic conditions and smoking status. The total score across all the domains ranges from 0–78, with higher scores representing healthier behaviors and lower risk factors for obesity. This scoring structure enables a detailed evaluation of the participants' lifestyle and environmental influences, facilitating targeted interventions and evidence-based recommendations.

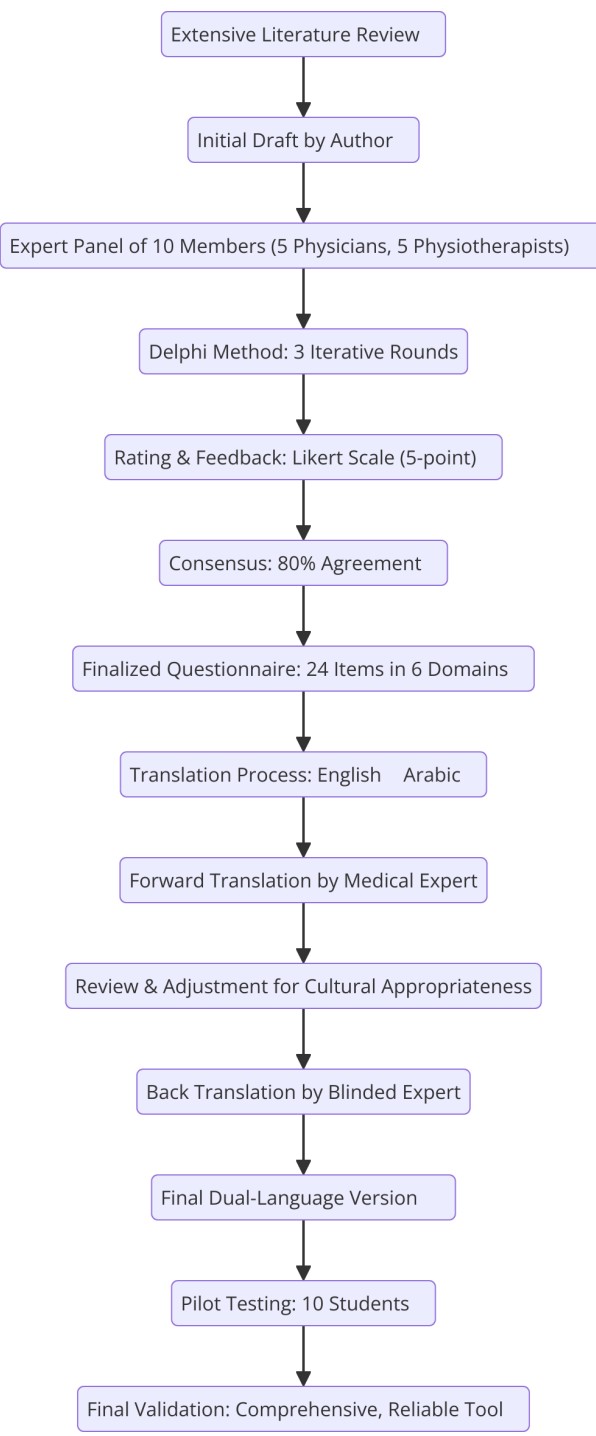

**Figure 2  Questionnaire development flowchart.**

The name of the scale was determined in consultation with five experts involved in tool development. By mutual agreement, all the experts approved the name Student Lifestyle and Obesity Risk Questionnaire (SLORQ), as it accurately reflects all the aspects being assessed and confirms its facial validity (File S1).

### Data collection

The participants were recruited for the study *via* convenience sampling. Those who met the inclusion criteria were included in the study. Permission and approval were sought from the concerned authorities prior to the college visits. The author of the paper personally recorded the participants' height and weight using a standard weighing scale and provided them with a hard copy of the questionnaire, which included this information. Students were able to either handover the completed questionnaire on the same day or email the scanned copy of completed questionnaire to the author. Objective measurements were conducted using a standardized digital weighing scale (Tanita BC-418MA) and a mobile stadiometer (Seca 213 Portable Stadiometer), with students barefoot and wearing minimal clothing. Each measurement was taken twice to ensure accuracy. Those willing to give their consent were given the option of selecting the language of the questionnaire they were most comfortable with (Arabic/English).

### Sample size

To determine the appropriate sample size for the study, the Raosoft sample size calculator was used with specific parameters tailored to the student population at Majmaah University. A population size of 12,888 was input, alongside a confidence level of 95% and a margin of error set at 3.3%, reflecting a commitment to precise and reliable results. The response distribution was assumed to be 50%, which is the standard for maximizing variability when the exact distribution is unknown. On the basis of these parameters, the recommended minimum sample size was 826 participants. To account for a potential dropout rate of 10%, the sample size was adjusted to ensure a final target of approximately 918 participants. This adjustment guarantees that the study will maintain sufficient statistical power and produce findings that are statistically significant and representative of the target population.

### Ethical considerations

Appropriate ethics committee approval (Approval Number: MUREC-Mar-21/COM-2024/10-6) was obtained from the Majmaah University Research Ethics Committee prior to the start of the study.

### Statistical analysis

Data analysis was performed *via* the IBM Statistical Package for the Social Sciences (SPSS) version 20.0 statistical software (IBM Corp., Armonk, NY, USA). Statistical analyses included normality tests using Shapiro–Wilk, descriptive statistics presented as mean $\pm$ standard deviation for normally distributed variables and median (interquartile range) for non-normally distributed variables, chi-square tests for categorical associations, and multivariable linear regression controlling for demographic and socioeconomic confounders. Reliability was assessed using Cronbach's alpha and intraclass correlation

coefficients (ICC). Principal component analysis (PCA) was performed to reduce the dimensionality of the data and identify underlying patterns in the variables. Prior to performing PCA, the suitability of the data for factor analysis was assessed using the Kaiser-Meyer-Olkin (KMO) measure of sampling adequacy and the Bartlett's test of sphericity. A KMO value greater than 0.60 and a significant Bartlett's test ($p < 0.05$) indicated that the data were suitable for PCA. A value of $\alpha \geq 0.70$ is generally considered acceptable for demonstrating internal consistency, with higher values indicating stronger reliability. Intrarater reliability was evaluated *via* the intraclass correlation coefficient (ICC) *via* a two-way mixed-effects model. Collinearity diagnostics revealed well-managed multicollinearity, with variance inflation factors (VIFs) ranging from 1.054–7.922. The Durbin–Watson statistic of 1.966 indicated no substantial autocorrelation in the residuals.

# RESULTS

## Demographic data

The prevalence of obesity through objective anthropometric measurements among college students at Majmaah University was 15% ($n = 136$), with the majority falling into the overweight BMI category at 40% ($n = 363$). However, there were no significant differences in sex, age, region, or college. A BMI above 30 kg/m$^2$ (obese) was associated with poorer health behaviors, as measured by the SLORQ (scores ranging from 24–38), whereas a BMI within the normal range was associated with better health behavior scores on the questionnaire (scores ranging from 55–65) (see Table 1).

## Correlations between BMI and health-related factors

Spearman's rank correlation analysis was conducted to assess the relationships between various health-related domains. A strong negative correlation was found between BMI and the dietary domain ($r = -0.912$, $p < .001$), followed by the weight management domain ($r = -0.877$, $p < .001$), physical activity domain ($r = -0.614$, $p$ lt .001), and sleep domain ($r = -0.567$, $p < .001$), indicating that higher BMI is associated with lower scores in these domains. Moderate positive correlations were observed between the physical activity domain and the dietary domain ($r = 0.637$, $p < .001$), sleep domain ($r = 0.717$, $p < .001$), and weight ($r = 0.732$, $p < .001$), suggesting that higher physical activity levels are associated with better outcomes in these areas. Similarly, a strong positive correlation was found between the dietary and weight management domain ($r = 0.896$, $p < .001$). Weak positive correlations were observed between the metabolic domain and the sleep domain ($r = 0.287$, $p < .001$), and the environmental domain showed moderate positive correlations with the weight management domain ($r = 0.664$, $p < .001$) and the SLORQ total score ($r = 0.812$, $p < .001$) (Fig. 3).

## Multiple linear regression analysis with BMI as the dependent variable

The linear regression analysis demonstrated a strong relationship between the predictors and the calculated BMI, with the model explaining 90.1% of the variance (adjusted $R^2 = 0.901$), indicating that a significant proportion of the variance was explained. The model was statistically significant (F(42, 864) = 196.819, $p < .001$) and had a root mean

Seyam (2025), *PeerJ*, DOI 10.7717/peerj.19556

Peer J

**Table 1** Demographic data.

| Variable (units) | Categories | BMI categories | | | | *p*-value |
|---|---|---|---|---|---|---|
| | | Under weight (>18 kg/m$^2$) | Normal (18.5–24.9 kg/m$^2$) | Over weight (25.0–29.9 kg/m$^2$) | Obese (<30 kg/m$^2$) | |
| Gender (n/%) | Male | 48 (48) | 167 (54.2) | 190 (52.3) | 68 (50) | 0.688[*] |
| | Female | 52 (52) | 141 (45.7) | 173 (47.65) | 68 (50) | |
| Age group (n/%) | 18–19 year old | 21 (21) | 69 (22.4) | 72 (19.8) | 31 (22.7) | |
| | 20–21 year old | 22 (22) | 52 (16.8) | 82 (22.5) | 30 (20.5) | |
| | 22–23 year old | 16 (16) | 63 (20.4) | 65 (17.9) | 28 (20.5) | 0.874[*] |
| | 24–25 year old | 22 (22) | 65 (21.1) | 74 (20.3) | 21 (15.4) | |
| | 25 and above year old | 19 (19) | 59 (19.1) | 70 (19.2) | 26 (19.1) | |
| Region (n/%) | Eastern | 17 (17) | 55 (17.8) | 81 (22.3) | 29 (21.3) | |
| | Middle | 13 (13) | 67 (21.7) | 84 (23.1) | 35 (25.7) | |
| | Northern | 26 (26) | 71 (23.0) | 61 (16.8) | 23 (19.9) | 0.145[*] |
| | Southern | 20 (20) | 52 (16.8) | 76 (20.9) | 26 (19.1) | |
| | Western | 24 (24) | 63 (20.4) | 61 (16.8) | 23 (16.9) | |
| College (n/%) | College of Applied Medical Sciences | 12 | 26 (8.4) | 31 (8.5) | 15 (11.02) | |
| | College of Business Administration | 9 (9) | 27 (8.7) | 34 (9.3) | 15 (11.02) | |
| | College of Dentistry at Zulfi | 15 (15) | 30 (9.7) | 34 (9.3) | 12 (8.8) | |
| | College of Education | 8 (8) | 32 (10.3) | 36 (9.9) | 13 (9.5) | |
| | College of Engineering | 14 (14) | 32 (10.3) | 37 (10.1) | 18 (13.2) | 0.24[*] |
| | College of Medicine | 17 (17) | 39 (12.6) | 39 (10.7) | 15 (11.02) | |
| | College of Nursing | 8 (8) | 42 (13.6) | 31 (8.5) | 8 (5.8) | |
| | College of Sciences | 7 (7) | 23 (7.4) | 33 (9.09) | 12 (8.8) | |
| | College of Sharia and Law | 7 (7) | 21 (11.6) | 47 (12.9) | 14 (10.2) | |
| | Computer Sciences and Information Technology College | 3 (3) | 36 (11.6) | 41 (11.2) | 14 (10.2) | |
| SLORQ Domain Score (m/SD) | Physical Activity Domain (Max score: 17) | 8.4 (1.3) | 14.07 (1.2) | 7.09 (0.9) | 6.6 (1.2) | 0.001[**] |
| | Dietary Domain (Max Score: 25) | 16.8 (0.6) | 15.09 (0.3) | 11.1 (0.4) | 6.08 (0.2) | 0.001[**] |
| | Sleep Domain (Max Score: 15) | 6.6 (0.7) | 10.3 (0.5) | 7.3 (0.6) | 4.5 (0.6) | 0.001[**] |
| | Metabolic Disorder Domain (Max Score: 3) | 2.8 (0.4) | 2.9 (0.1) | 2.9 (0.1) | 2.6 (0.6) | 0.001[**] |
| | Weight management Domain (Max Score: 20) | 9.7 (0.4) | 9.8 (0.4) | 6.07 (0.3) | 4.4 (0.7) | 0.001[**] |
| | Environment factor domain (Max Score: 15) | 8.7 (1.04) | 11.5 (2.5) | 8.9 (1.06) | 7.1 (1.1) | 0.001[**] |

**Seyam (2025), *PeerJ*, DOI 10.7717/peerj.19556**

**Table 1** (*continued*)

| Variable (units) | Categories | BMI categories | | | | *p*-value |
| --- | --- | --- | --- | --- | --- | --- |
| | | Under weight ($>18$ kg/m$^2$) | Normal ($18.5–24.9$ kg/m$^2$) | Over weight ($25.0–29.9$ kg/m$^2$) | Obese ($<30$ kg/m$^2$) | |
| SLORQ Total score (m/SD) | Sum of six domains (Max Score:104) | 53.2 (1.9) | 63.9 (2.07) | 43.5 (1.3) | 31.5 (1.6) | 0.001[**] |
| Body Mass Index (n/%) | categories | 100 (11.0) | 308 (34.0) | 363 (40.0) | 136 (15.0) | ND |

**Notes.**

Chi-Square Test, indicated by * for statistically significant differences ($p < .05$). Column comparisons were conducted with the *Z*-test, applying Bonferroni adjustment for multiple comparisons. Mean values were compared using ANOVA, with statistically significant differences denoted by **; ND, not determined: SLORQ, Student Lifestyle and Obesity Risk Questionnaire.

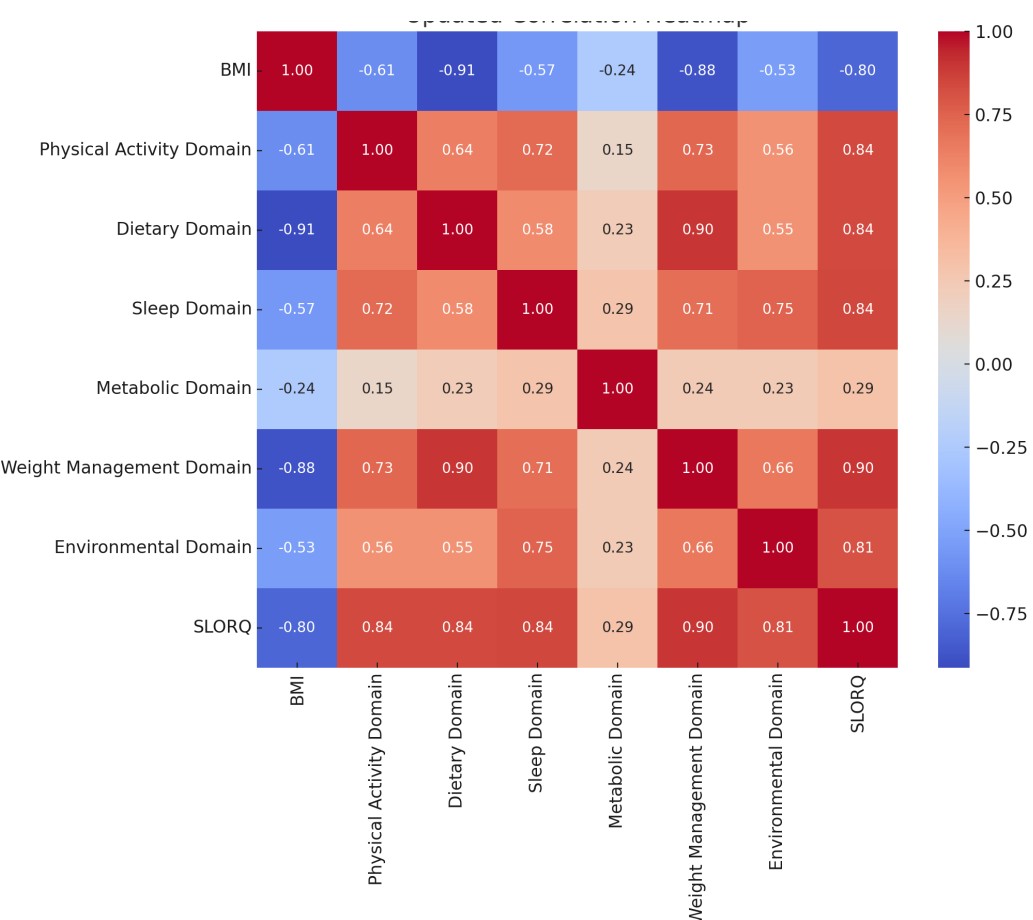

**Figure 3   Correlation between variables.**

square error (RMSE) of 1.54. The significant contributors included the dietary habits domain related to the consumption of fruits and vegetables ($\beta = -0.322$, $t = -4.465$, $p < .001$) and the dietary habits domain related to the primary source of protein ($\beta = -0.615$, $t = -12.304$, $p < .001$), both of which strongly negatively associated with BMI, as did the metabolic health domain related to family history of diabetes or metabolic syndrome ($\beta = -0.032$, $t = -2.329$, $p = .02$), with a smaller effect size. Near-significant predictors included physical activity related to the frequency of physical activity ($\beta = 0.03$, $t = 1.854$, $p = .064$), the sleep patterns domain related to hours of sleep ($\beta = -0.041$, $t = -1.862$, $p = .063$), the weight management domain related to prediabetes ($\beta = -0.063$, $t = -1.832$, $p = .067$), and the environmental and family factor domain related to smoking status ($\beta = -0.039$, $t = -1.743$, $p = .082$), suggesting trends that warrant further exploration. Other factors, such as physical activity related to the type of physical activity ($\beta = -0.038$, $t = -1.026$, $p = .305$) and dietary habits related to counting calories ($\beta = -0.043$, $t = -0.519$, $p = .604$), were not significant (see Table 2).

**Table 2  Regression table with BMI as dependent variable.**

| Variable | Unstandardized coefficient (B) | Standard error | Standardized coefficient (Beta) | $t$ | $p$-value | 95% CI lower | 95% CI upper | Tolerance | VIF |
|---|---|---|---|---|---|---|---|---|---|
| (Intercept) ($H_0$) | 25.084 | 0.162 | – | 154.515 | <0.001 | 24.766 | 25.403 | – | – |
| (Intercept) ($H_1$) | 35.929 | 1.296 | – | 27.715 | <0.001 | 33.385 | 38.474 | – | – |
| PA2 | −0.589 | 0.101 | −0.125 | −5.835 | <0.001 | −0.788 | −0.391 | 0.477 | 2.096 |
| PA3 | −0.15 | 0.128 | −0.033 | −1.173 | 0.241 | −0.4 | 0.101 | 0.279 | 3.59 |
| PA4 | −0.21 | 0.106 | −0.03 | −1.992 | 0.047 | −0.418 | −0.003 | 0.987 | 1.013 |
| DH3 | −1.383 | 0.183 | −0.221 | −7.575 | <0.001 | −1.742 | −1.025 | 0.258 | 3.876 |
| SP1 | −0.484 | 0.129 | −0.106 | −3.766 | <0.001 | −0.736 | −0.232 | 0.277 | 3.616 |
| SP2 | 0.774 | 0.196 | 0.107 | 3.954 | <0.001 | 0.39 | 1.158 | 0.303 | 3.305 |
| MH1 | 2.52 | 0.544 | 0.081 | 4.636 | <0.001 | 1.453 | 3.587 | 0.72 | 1.388 |
| MH2 | −2.479 | 0.388 | −0.116 | −6.387 | <0.001 | −3.241 | −1.717 | 0.67 | 1.492 |
| MH3 | −1.519 | 0.536 | −0.043 | −2.836 | 0.005 | −2.57 | −0.468 | 0.943 | 1.061 |
| WM3 | 0.574 | 0.462 | 0.02 | 1.243 | 0.214 | −0.332 | 1.479 | 0.826 | 1.211 |
| WM4 | −3.873 | 0.203 | −0.562 | −19.054 | <0.001 | −4.271 | −3.474 | 0.253 | 3.948 |
| EF1 | 3.45 | 0.248 | 0.333 | 13.898 | <0.001 | 2.963 | 3.938 | 0.383 | 2.61 |
| EF2 | −0.087 | 0.051 | −0.03 | −1.703 | 0.089 | −0.187 | 0.013 | 0.69 | 1.449 |
| EF3 | −0.913 | 0.232 | −0.091 | −3.929 | <0.001 | −1.369 | −0.457 | 0.409 | 2.448 |
| EF4 | −0.026 | 0.491 | $-8.386 \times 10^{-4}$ | −0.052 | 0.959 | −0.99 | 0.939 | 0.846 | 1.182 |

**Notes.**

An intercept was included in the regression model to account for the baseline level of the dependent variable when all predictors are at zero.

PA1-PA4, Physical Activity Domain (Types, Frequency, Intensity, On-campus activity); DH1-DH5, Dietary Habits Domain (Fruits/Vegetables, Protein sources, Fast food, Sugary foods, Calorie balance); SP1-SP3, Sleep Patterns Domain (Hours, Well-rested feeling, Difficulty sleeping); MH1-MH3, Metabolic Health Domain (Insulin resistance, Family history, Thyroid disorder); WM1-WM4, Weight Management Domain (Weight fluctuation, Monitoring, Interventions, Support system); EF1-EF4, Environmental and Family Factors Domain (Nutrition awareness, Campus amenities, Food availability, Chronic conditions).

## Internal consistency and principal component analysis of the questionnaire

The Cronbach's alpha for the 24 items was 0.915, indicating excellent internal consistency. Additionally, a two-way mixed-effects model was employed to calculate the ICC, with the assumption that people effects were random and that measure effects were fixed. The single-measure ICC was .319 (95% CI [0.298–0.342]), and the average-measure ICC was .915 (95% CI [0.907–0.923]). Both ICC values were statistically significant, F(906, 19,932) = 11.788, $p < .001$, demonstrating strong reliability for the aggregated measures.

To check for redundancy of the items in the scale due to high internal consistency, PCA was performed to identify the overlapping constructs (File S1). The PCA identified five components explaining 75.6% of the variance. Items with high uniqueness or low contribution, such as weight management item 3 (uniqueness = 0.735), environmental factor item 4 (uniqueness = 0.772), and metabolic health item 3 (uniqueness = 0.915), should be re-evaluated for inclusion in the scale. Additionally, items with cross-loadings, such as Environmental Factors Item 1 (RC1 = 0.787, RC2 = 0.507, RC3 = −0.727), may need refinement to address multidimensionality and improve scale clarity.

### Intra-rater reliability

The intra-rater reliability of the scale was assessed among 46 participants after an interval of one week *via* the ICC *via* a two-way mixed-effects model, where people effects were random and measures effects were fixed. The analysis demonstrated excellent consistency within the raters, with a single-measures ICC of 0.998 and an average-measures ICC of 0.999.

## DISCUSSION

This study found that the prevalence of obesity among Majmaah University students was 15%, with dietary patterns, sleep duration, and physical activity being the strongest predictors of BMI. Importantly, no significant differences emerged based on gender or region. The lack of significant gender differences in obesity prevalence in this study contrasts with previous research findings of higher obesity rates among females in SA (*Salem et al., 2022*). This difference may reflect recent shifts in cultural attitudes toward health awareness and physical activity participation at Majmaah University, potentially reducing gender-related disparities. Additionally, self-selection bias from voluntary participation in the study might have influenced these results. These findings are consistent with global trends showing increasing rates of overweight and obesity among college students, attributed to unhealthy lifestyle behaviors, dietary patterns, and physical inactivity (*Ejtahed et al., 2024*; *Shafiee et al., 2024*).

### Development of a multidimensional questionnaire

The motivation to develop a new questionnaire arose from the lack of a comprehensive tool that includes all the domains of obesity risk factors that are available in both English and Arabic languages to be used among college students. A number of questionnaires are available to record obesity among college students. However, most of them are either not available in the Arabic language or are not able to cover all the risk factors associated with obesity. These questionnaires include the Weight and Lifestyle Inventory (*Fink-Miller & Rigby, 2017*), College Student Obesity Questionnaire (*Wang et al., 2022*), National College Health Assessment (*American College Health Association, 2009*), Youth Risk Behavior Surveillance System (*Sussman et al., 2002*), and Lifestyle and Health Behavior Questionnaire (*Lakerveld et al., 2011*). These questionnaires often face limitations such as a lack of cultural adaptation, insufficient focus on college-specific factors, inadequate validation across diverse populations, and the omission of mental health and environmental influences on obesity.

### Obesity prevalence findings

The current study revealed a prevalence of 40% college students overweight. Similarly, a cross-sectional study in the southwestern part of SA among 540 medical college students revealed 14.6% obesity with 21% overweight, which was half of what we found in our study. The higher prevalence of overweight in our study may be attributed to differences in sample demographics, regional or cultural factors, data collection methods, diagnostic criteria, lifestyle behaviors, or temporal changes in obesity trends. Further it may be due

to psychological barriers to being evaluated because of being overweight (*Mahfouz et al., 2024*). Conversely, a health science college in the Taif region of SA reported an overweight rate of 25.9% (*Hamam et al., 2017*). Similarly, a college in Alkharj, SA, among 1,019 students, reported 32.3% students overweight (*Al-Ghamdi et al., 2018*).

## Gender and regional differences

Our results revealed no significant differences in BMI categories based on sex. In contrast, a study involving young Saudi adolescents (14–19 years old) reported a lower prevalence of obesity in females (14%) than in males (24.1%) (*Al-Hazzaa et al., 2014*). This is intriguing, as many studies have reported gender disparities in the prevalence of obesity, often citing higher rates among females due to biological, hormonal, and behavioral factors (*Leeners et al., 2017*; *Pucci et al., 2017*; *Ciarambino et al., 2022*). For example, a study revealed that Saudi females tend to engage in less physical activity than their male counterparts do, potentially predisposing them to higher obesity rates (*Al-Hazzaa et al., 2014*). The absence of such differences in our study might be attributed to the changing cultural and academic environment at Majmaah University, where students are increasingly becoming more health-conscious, regardless of gender, as observed in previous studies (*Alsulami et al., 2023*).

Regional differences in BMI categories were not statistically significant; however, students from the middle and eastern regions of Saudi Arabia exhibited marginally higher obesity rates in our study. Similarly, a systematic review reported the highest prevalence of obesity in the eastern region (*Salem et al., 2022*). This finding may reflect regional dietary patterns, socioeconomic disparities, or cultural practices among students.

## Behavioral risk factors (diet, physical activity)

The physical activity scores were markedly lower among obese students, emphasizing the critical role of exercise in weight management. This finding aligns with the global literature, which consistently identifies physical inactivity as a leading risk factor for obesity (*Silveira et al., 2022*). Despite the growing emphasis on campus-based physical activity initiatives, our study underscores the need for targeted interventions to promote regular and diverse physical activities among students.

Dietary domain scores were significantly lower in obese students, highlighting poor dietary practices such as excessive fast food consumption and inadequate intake of fruits and vegetables. These results resonate with those of studies that reported similar dietary trends among Saudi youth (*Musaiger, Hassan & Obeid, 2011*). The cultural preference for calorie-dense traditional foods and the rising popularity of Western fast-food chains may exacerbate these dietary behaviors (*Mahmood et al., 2024*). Educational campaigns focusing on nutrition literacy could play a pivotal role in addressing this issue. Previous research in SA has highlighted regional variations in obesity, which are often linked to differences in urbanization, access to physical activity facilities, and traditional food consumption patterns (*Al-Nozha et al., 2005*).

## Sleep and metabolic health

The sleep domain revealed significantly poorer sleep behaviors among obese students, with shorter sleep durations and poorer sleep quality. The evidence from previous studies suggests a strong bidirectional relationship between sleep and obesity, with sleep deprivation leading to hormonal imbalances that promote weight gain (*Chaput et al., 2016*). Universities could consider implementing programs to raise awareness about the importance of healthy sleep habits and their impact on overall health. Scores in the weight management domain were significantly lower among obese students, indicating inadequate monitoring of weight and inconsistent efforts at weight control. This aligns with findings from *Al-Rethaiaa, Fahmy & Al-Shwaiyat (2010)*, who reported that Saudi youth often lack regular weight monitoring practices. Enhancing access to weight management resources and counseling services on campuses could encourage proactive approaches to weight control. Environmental and family factors also emerged as significant determinants, with lower scores correlating with higher BMI categories. The availability of unhealthy food options on campuses and limited awareness of nutritional content were notable barriers. A family history of metabolic disorders further compounds the risk, emphasizing the need for a multifaceted approach that addresses both individual behaviors and environmental influences.

## Strengths and limitations

The strengths of this study include its large sample size, comprehensive assessment of obesity-related domains, and the use of validated and culturally adapted tools. However, this study utilized convenience sampling, limiting generalizability to the broader university population. Additionally, self-report measures, although validated, may have introduced recall biases. Future research should use probability sampling methods, longitudinal study designs, and complementary measures (*e.g.*, bioelectrical impedance analysis) to enhance representativeness and accuracy of obesity assessments. Additionally, the cross-sectional design precludes causal inferences, while SLORQ provides comprehensive insights into obesity-related behaviors and environmental factors and social desirability biases. Therefore, future studies should incorporate objective validations, such as accelerometer for physical activity and direct dietary assessments, to strengthen the tool's validity. Although demographic variables were controlled for in the multivariate analysis, socioeconomic status and errors in reporting dietary intake were not measured, which may have influenced the results. BMI is widely used to classify obesity, it has inherent limitations due to its inability to differentiate between fat and muscle mass. This constraint emphasizes the need for complementary assessments, such as bioelectrical impedance analysis, to precisely evaluate body composition (*Jakubiak et al., 2024*).

## Policy implications

Future research should explore longitudinal designs to examine trends in weight changes during college years. Interventions targeting specific risk factors identified in this study should be developed and evaluated for their effectiveness. To combat obesity effectively, Majmaah University could implement specific policy interventions, such as improving

the nutritional quality of campus meals, providing incentives for student engagement in structured physical activities, enhancing the campus environment to encourage active lifestyles, and organizing educational programs emphasizing the importance of adequate sleep and nutrition.

## CONCLUSIONS

This study successfully addressed two primary objectives: establishing the prevalence and identifying associated risk factors for obesity among university students and validating the newly developed SLORQ. SLORQ emerged as a reliable, culturally adapted tool capable of informing targeted campus-based obesity interventions. Universities are encouraged to leverage these findings by creating supportive environments that foster healthier dietary practices, regular physical activity, and sufficient sleep, thereby significantly reducing obesity risks among students.

## ACKNOWLEDGEMENTS

I would like to thank all the experts for their contribution in developing the questionnaire for this study and the students who participated in this study.

### Funding

This work was funded by the Deanship of Postgraduate Studies and Scientific Research at Majmaah University through project no. R-2025-1788. The funders had no role in study design, data collection and analysis, decision to publish, or preparation of the manuscript.

### Grant Disclosures

The following grant information was disclosed by the author:
Deanship of Postgraduate Studies and Scientific Research at Majmaah University: R-2025-1788.

### Competing Interests

The author declares that they have no competing interests.

### Author Contributions

- Mohamed K. Seyam conceived and designed the experiments, performed the experiments, analyzed the data, prepared figures and/or tables, authored or reviewed drafts of the article, and approved the final draft.

### Human Ethics

The following information was supplied relating to ethical approvals (i.e., approving body and any reference numbers):

This research was approved by Majmaah University Research Ethics committee (MUREC-Mar-21/COM-2024/10-6).

## Data Availability

The raw data is available in the Supplementary File.

## Supplemental Information

Supplemental information for this article can be found online at http://dx.doi.org/10.7717/peerj.19556#supplemental-information.

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
