# Peer review of "Prevalence of obesity and associated risk factors among university students using the newly developed Student Lifestyle and Obesity Risk Questionnaire (SLORQ): a cross-sectional study"

_PeerJ, doi:10.7717/peerj.19556_

## Round 0.1 · original submission · Major Revisions

Your manuscript requires major revisions.
Here are some of the principal issues mentioned by reviewers:

Study Focus: It is unclear whether you are purely examining obesity prevalence/risk factors or also introducing/validating a new questionnaire. If a new tool is central, clarify and reflect that throughout the title, introduction, methods, and discussion.

Methods Clarification:
Anthropometrics: Specify if height/weight were self-reported or objectively measured (and how).
Questionnaire: Consolidate all information about its design, domains, scoring, and validation into one clear subsection.
Statistics: Clearly describe how you tested normality, controlled for confounders, and performed reliability/validity checks (if validating a new tool).

Results: Avoid repetition and clearly distinguish findings from self-reported vs. objectively measured data. Present results in a logical order, and ensure references to the new tool are accurate (if relevant).

Discussion:
Align with the aims (e.g., new tool validation or prevalence/risk-factor analysis).
Support statements with appropriate references.
Highlight practical implications for university health programs and address the limitations of using BMI alone.

Language & Structure:
The manuscript needs significant language revision, removal of redundancies, and consistent use of abbreviations.
Expand the limitations section (e.g., BMI vs. body composition, self-report data, sampling approach).

**Language Note:** The review process has identified that the English language must be improved. PeerJ can provide language editing services - please contact us at [email protected] for pricing (be sure to provide your manuscript number and title). Alternatively, you should make your own arrangements to improve the language quality and provide details in your response letter. – PeerJ Staff

·

Basic reporting

This is a revision of the manuscript “Prevalence of obesity and associated risk factors among university students: A cross-sectional study” submitted to PeerJ.

Abstract:
The final sentence of the Background and the Objective section is very similar. Please, consider changing the background section, replacing the current sentence with another to highlight why this paper is needed (literature gap, etc.).
The Methods section is unclear whether the students were objectively measured or if the weight and height were self-reported. Please, improve the text.
Overall, the abstract is informative and reflects all the important information from the research.

Introduction:
Line 49: Please, define any abbreviation in its first use. The text should be “…attributed to high body mass index (BMI)…”
Lines 51 to 53 can be: “Recent studies estimate that by 2040, over 2.26 million new cases of Type 2 diabetes mellitus, chronic liver diseases, and liver cancer in South America will be attributed to obesity in young adults.” In Lines 56-59, the author presents a study conducted among students from 22 countries, but by the end, he inserts another study. Please, separate the two sentences. Overall, the text needs to go through an English revision. And the flow can be improved, for instance, the Introduction starts by showing the burden of obesity worldwide and in SA, and it follows with the specific sample of university students. However, in line 61, the author inserts the information about the economic burden. This information has no place in its current position, since it is not specific to young adults.
Line 55: may need a reference showing how life habits can be adopted at this stage in life and persist in adulthood.
Line 59: There is no need to define (again, first used in line 50) the Saudi Arabia abbreviation.
Line 74/75: As it is, it seems that the author is going to test/evaluate/validate a new tool, but this is not the study aim. Please improve the text by clarifying the objective of the current manuscript.

Materials & Methods:
Line 90: were the questionnaires delivered and completed in paper or online? Please, provide more details. I found this information later in lines 158-160. Overall, I think the text is not clear because its structure is not correct. This can be improved.
Line 101: I’m still in doubt if the validation of the questionnaire is one of the points of the current manuscript…The author provides an extensive explanation for the development of the questionnaire. Please clearly state the objective of the paper in both the abstract and the introduction section. Also, the title does not reflect that the manuscript is validating a new tool. Please, make the appropriate changes.
Line 105: “The initial draft of the questionnaire was formulated by the author of this paper, and later it the circulated among experts”; please correct the English.
Line 110: “finalized questionnaire of 8 for demographic data and 24 items distributed”; please modify to “8 demographic items and 24 items…”
Some information is repeated throughout the text, which demands a revision. For instance, lines 128-130 repeat information previously delivered (in the previous paragraph) regarding the “six domains”, the “bilingual”, and “available in both English and Arabic”.
Please clarify what was measured in the questionnaire by asking students about “physical activities” (lines 133-136). Does the item include any activity that makes our body move and involves energy expenditure? Does it account for organized and non-organized activities? This needs to be clarified.
Line 147: again, the repetition of information, with the author mentioning the maximum score range of 78 (which was previously mentioned in line 124). I do believe that line 147 is the appropriate place for this information. Please, modify the text to avoid repetition of ideas and information, to shorten the text and clarify for the reader.
Line 162: The anthropometric data collection is not clear. The questionnaire was self-reported and had a section for putting in the height and weight. But the author also states in the mentioned line that these measures were objectively collected. Does this mean that the project collected subjective and objective data on weight and height? Please, be more specific throughout the text. Also, provide more information regarding the participants, including: were they barefoot?

Results:
English revision throughout the text. An example, line 197: “While there were no significant differences observed in gender, age, regions, or colleges.” The use of the word “while” presumes that there will be a comparison, but this does not happen in the current sentence.
Line 199/200: the abbreviation of the questionnaire name was previously used, and does not make sense to put the full name at this point. Please modify.
The section “Internal consistency of the questionnaire” mixes methodology (i.e., statistical tests) with results (i.e., actual results of the statistical tests). Again, this reflects the poor structure of the manuscript.
Structural problems are also shown in sections Demographic data and Correlations between BMI and health-related factors. The first makes an introduction of the factors related to the BMI, and I doubt that this is the right place to provide that information.
Line 237: This is a cross-sectional study, so please avoid using terms that may suggest impact or effect instead of a correlation/association, as shown in “higher physical activity levels are associated with improvements in these areas”. Replace the word “improvements”.

Discussion:
Line 266-268: needs a reference showing the increasing rates of obesity among college students attributed to unhealthy lifestyles.
This section continues to present repetition of ideas, such as line 261 and line 283.
Line 284/285: the obesity rate is very similar, only the overweight rate is half of the one found in this paper. Improve the sentence. Also, the explanation that the overweight difference (but not obesity) may be because the sample of the other study was gathered from just one college does not make much sense. Can the author explain his thought?
Line 302: “…irrespective of gender are self-conscious about their health.” – Is this a finding/conclusion of the current study? If not, insert a reference or change the sentence to make it as an explanation for the current results.
Line 304/305: “…reported marginal high obesity prevalence.” – Again, I have no idea which obesity prevalence the author is reporting: the self-reporting data in the questionnaire? Or the values that came from the objectively measured?
Line 316: “…studies by (Musaiger, Hassan & Obeid, 2011)…” – This is not the correct way to make a reference.
Overall, the discussion can be greatly improved by providing more critical discussion regarding the direct and indirect associations between the variables. Also, the conclusion is not very informative, still ignoring the validation of the tool, which took a great part of the manuscript, but is ignored is several major points such as the title, the aim, and the conclusion. Also, I would advise more “go home message” in the conclusion, providing more information on how the university campus can be more “supportive” in the multiple points that were related to the obesity prevalence.

Experimental design

-

Validity of the findings

-

Additional comments

Overall, the manuscript presents an interesting and valuable contribution by introducing a new tool to evaluate obesity rates among students and the related variables. This is particularly important given that students represent a vulnerable group, and few studies have analyzed the interaction between multiple contributing factors.

However, the manuscript does not effectively demonstrate the interactions between these variables. Additionally, several sections contain repetitive ideas and results, which could be streamlined to improve clarity and readability. Enhancing these aspects will strengthen the manuscript’s impact and coherence.

Reviewer 2 ·

Basic reporting

Below, I provide detailed feedback, along with specific recommendations for improvement.

Title:
The study introduces a new obesity risk assessment tool (SLORQ), but the title does not indicate this methodological contribution.
If SLORQ is a key innovation, the title could be "Prevalence of Obesity and Associated Risk Factors among University Students in Saudi Arabia Using a Newly Developed Obesity Risk Assessment Tool: A Cross-Sectional Study."

Abstract:
The abstract moves abruptly from methods to results, making it less fluid. Use a brief linking sentence between methods and results (e.g., "Following data collection, statistical analyses revealed...").
Add a statement confirming SLORQ's reliability and internal consistency (Cronbach’s α = 0.915, ICC = 0.999).
Add a sentence in the results:
"No significant differences were observed based on gender, age, or region."
The adjusted R² = 0.901 is reported, but this may be too technical for an abstract.
Suggestion: Instead of focusing on the statistical value, state the practical implication:
"Diet, sleep, and physical activity accounted for most of the variance in BMI, highlighting their importance in obesity prevention."

Introduction:
While the study aims to examine obesity risk factors, it does not explicitly highlight the gap in existing literature.
There is mention of existing questionnaires, but no strong justification for developing a new tool (SLORQ) beyond cultural relevance. Clearly define what existing studies have missed and why the SLORQ is needed beyond existing tools.
The last paragraph introduces the potential application of findings but lacks a clear statement on how the study contributes to policy, intervention development, or public health strategies.

Methods:
Address the limitations of convenience sampling and discuss how this might affect representativeness.
The description of the SLORQ domains is scattered across different sections, making it difficult to track the structure of the tool. Present all SLORQ domains and their item counts. (For example, the physical activity domain consists of 5 items with a total score range between (x-x ).
While multivariable regression was used, the study does not mention how confounding variables (e.g., socioeconomic status, dietary intake measurement errors) were controlled. Specify whether confounders were considered and if adjustments were made during analysis.
Inconsistencies in Data Collection Description: The time of measurement (before breakfast) is noted, but dietary intake assessment was self-reported without objective verification. Discuss potential recall bias in self-reported dietary and activity data.
The statistical methods (e.g., regression models, PCA) are placed after the data collection section, which disrupts the logical flow. Move the statistical methods section to the end of the Methods, following data collection and ethical considerations.

Discussion

Start with a summary of the key findings (e.g., “This study found that the prevalence of obesity among Majmaah University students was 15%, with dietary patterns, sleep duration, and physical activity being the strongest predictors of BMI.”).

Some sections lack smooth transitions, making the discussion harder to follow.
Use a structured approach to present findings logically:
1. Obesity prevalence findings → 2. Behavioral risk factors (diet, physical activity) → 3. Sleep and metabolic health → 4. Gender and regional differences → 5. Policy implications.

The study concludes that health promotion programs are needed, but it does not provide specific recommendations for university policies. Suggest targeted interventions (e.g., campus meal modifications, physical activity incentives, structured wellness programs).

Provide a stronger rationale for why no gender difference was observed, considering factors like changing cultural norms, university environment, or selection bias.

Some studies reported a higher prevalence among male adolescents than female adolescents. [doi: 10.4103/sjg.SJG_617_18]

The SLORQ questionnaire is central to the study, but no potential limitations of the tool are discussed.
Recommendation: Address limitations of the SLORQ (e.g., reliance on self-reported data, need for objective validation).

Experimental design

-

Validity of the findings

-

Reviewer 3 ·

Basic reporting

I received for review an original research article entitled "Prevalence of obesity and associated risk factors among university students: A cross-sectional study", prepared by Mohamed K Seyam, which was submitted to the PeerJ. Obesity and its complications are one of the most important challenges for public health in the modern world. Obesity predisposes to the development of metabolic syndrome, which is a set of disorders contributing to increased cardiovascular risk and the development of overt cardiovascular disease. Cardiovascular diseases, especially in the course of atherosclerosis, are the main cause of morbidity and mortality in many countries of the world. Understanding the factors that determine its development, including the factors that determine the development of obesity and its complications, is therefore of key importance. In my opinion, the subject matter is therefore extremely important. In my opinion, the manuscript presents a certain scientific value. However, some significant changes are necessary, which may contribute to increasing the value and attractiveness of the presented manuscript.
1) The purpose of the work must be precisely defined. You cannot write that "This study aims to provide valuable insights into current prevalence and the specific risk factors and behaviors contributing to obesity.", because such a sentence does not provide any information. What question did the authors want to answer by conducting the described scientific research? Was the purpose of the study only to collect data or also to develop a new research tool?
2) In the Material and Methods there is “52.1% male and 46.9% female”. Men and women together make up 99% of this information.
3) The description of the statistical analysis methodology is too poor. How was the conformity of the distribution to the normal distribution tested? Were the mean and standard deviation used for variables with a normal distribution, and the median and interquartile range for other variables?
4) All aspects of the presented results should be carefully discussed. Meanwhile, the discussion in the current version refers mainly to prevalence and gender differences in BMI categories.
5) In my opinion, it is worth mentioning the significant limitations of BMI. BMI is a derivative of body mass and height only. It does not refer to body composition, including the content and distribution of adipose tissue. A valuable supplement is the analysis of body composition using the bioelectrical impedance method. I suggest referring to the following publication: 10.3390/medicina60071080.
6) The article is written rather carelessly in terms of editing, language, and style. Please review the entire manuscript for minor editorial, language, editorial, and style corrections.

Experimental design

-

Validity of the findings

-

---

## Round 0.2 · Minor Revisions

Thank you for the careful revision of the reviewers' commentaries.

While all the comments were addressed by the authors, there are some minor issues that I have listed in the attached file. Please review carefully and correct them.

Reviewer 3 ·

Basic reporting

I received for review a revised version of the original research article entitled "Prevalence of obesity and associated risk factors among university students: A cross-sectional study" (currently: "Prevalence of Obesity and Associated Risk Factors among University Students Using the Newly Developed Student Lifestyle and Obesity Risk Questionnaire (SLORQ): A Cross-Sectional Study"), prepared by Mohamed K Seyam, which was submitted to the PeerJ. In my opinion, the paper has been significantly improved. I have no further critical comment. Thank you for the invitation to prepare a review report of the presented manuscript. Congratulations to the Author and best wished for your further scientific work.

Experimental design

No additional comments.

Validity of the findings

No additional comments.

Additional comments

No additional comments.

---

## Round 0.3 · accepted · Accept

Thanks to the authors for addressing all the proposed changes by reviewers.
The manuscript is ready for publication.